# Comprehensive Understanding of Aluminosilicate Phosphate Geopolymers: A Critical Review

**DOI:** 10.3390/ma15175961

**Published:** 2022-08-29

**Authors:** Shanliang Ma, Zengqi Zhang, Xiaoming Liu

**Affiliations:** 1State Key Laboratory of Advanced Metallurgy, University of Science and Technology Beijing, Beijing 100083, China; 2School of Metallurgical and Ecological Engineering, University of Science and Technology Beijing, Beijing 100083, China

**Keywords:** aluminosilicate phosphate geopolymers, synthetic process, reaction mechanism, mechanical property, modification

## Abstract

Aluminosilicate phosphate (ASP) geopolymers are a new kind of green cementitious materials synthesized from aluminosilicate precursors and acidic activators (phosphoric acid or phosphate), which have received extensive attention from researchers because of their excellent and unique characteristics. The current investigation indicates that ASP geopolymers have the characteristics of a low-carbon synthesis process, high mechanical properties (e.g., the highest compressive strength can reach 146 MPa), a strong heat resistance (e.g., withstanding a high temperature of 1500 °C), and excellent dielectric properties. These excellent properties make them have broad application prospects in the fields of new building materials, coating materials, insulating materials, and heavy metal curing. Based on the research findings of approximately 85 relevant literatures on ASP geopolymers in past decades, this paper focuses on the latest research progress of ASP geopolymers from the perspectives of synthesis processes, performances, modifications, and application developments. In addition, this study summarizes the key problems existing in the current research of ASP geopolymers and suggests their possible applications in the future, which will help to provide directions for further research activities of relevant researchers.

## 1. Introduction

Geopolymers are a kind of inorganic polymer material with a three-dimensional network structure from amorphous to semicrystalline [1]. Initially, geopolymers were strictly defined as alkali-aluminosilicate (AAS) geopolymers produced by the reaction of aluminosilicate precursors with an alkali, which is a three-dimensional network gel material formed by the depolymerization, polycondensation, and gel network process of natural active aluminosilicate minerals (such as metakaolin and volcanic pumice dust) or aluminosilicate industrial wastes (such as blast furnace slag) in strong alkali solutions [2,3,4,5]. Geopolymers are superior to Portland cement in strength, corrosion resistance, and thermal stability by the special formation method [6]. In particular, geopolymers are a kind of low-carbon environmental material, and the CO_2_ emissions in its synthesis process are only approximately 20% of the emissions from the production of Portland cement [7]. The raw materials, reaction mechanism, modification, and various properties of AAS geopolymers have been widely studied and explored [8,9,10,11], and some research results have been successfully applied in engineering [12,13].

Unlike AAS geopolymers, although there was evidence proving that phosphate geopolymers had been discovered and used for a long time (even dating back to ancient Egypt [14]), there was a lack of systematic understanding of it. Scholars had a comprehensive understanding of the phosphate geopolymer and gradually began to carry out related systematic research until Argonne National Laboratory put forward the concept of “phosphate chemically bonded ceramics” and the related research work of aluminosilicate phosphorate geopolymers carried out by Cao et al. [15,16]. Davidovits also officially wrote phosphate geopolymers into his authoritative monograph on geopolymer chemistry and application in 2011 [2]. Aluminosilicate phosphate (ASP) geopolymers refer in particular to phosphate geopolymers formed by the reaction of aluminosilicate precursors with acidic activators (phosphoric acid or phosphate).

Prior studies have shown that ASP geopolymers have excellent mechanical properties [17], heat resistance [18], dielectric properties [19], etc. The structure of ASP geopolymers is similar to AAS geopolymers; aluminum or silicate in its network structure is partially or completely replaced by phosphorus, and the basic units of ASP geopolymers are –Si–O–Al–O–P–, –Si–O–P–O–Al–, or –Al–O–P– [20,21]. The reaction process of aluminosilicate precursor forming geopolymers in alkaline and acidic environments is shown in Figure 1. From the perspective of topology, phosphorus oxygen tetrahedron is positively monovalent when it enters the network structure of geopolymers, which can balance the negative charge of aluminum oxygen tetrahedron and ensure the stability of the ASP geopolymer structure [16]. The reaction of AAS geopolymers takes place in a strong alkali environment, while the geopolymerization of ASP geopolymers takes place in an acidic solution. The main reaction and structural empirical formula are shown in Equations (1) and (2), respectively [3].
(1)$${\mathrm{Al}}_{2}O_{3}{\cdot 2\mathrm{SiO}}_{2}\;{+2H}_{3}{\mathrm{PO}}_{4}\;\to {\;\mathrm{Al}}_{2}{\mathrm{Si}}_{2}P_{2}O_{X}{\cdot H}_{2}O\;\left(\mathrm{amorphous}\;\mathrm{gel}\right)\;$$
(2)$${[-{\left(\mathrm{Si}-O\right)}_{Z}-\mathrm{Al}-O-P]}_{n}{\cdot \mathrm{mH}}_{2}O$$

The number of papers and patents published on ASP geopolymers were counted by using the keywords “phosphate” and “geopolymer”. As shown in Figure 2, the research of ASP geopolymers has been greatly developed in recent years, which indicates that more and more researchers begin to pay attention to the new geopolymer materials. Based on the research findings of nearly 85 relevant literatures on ASP geopolymers in past decades, this paper focuses on the latest research progress of ASP geopolymers from the perspectives of synthesis processes, performance, modification, and applications development. In addition, this study summarizes the key problems existing in the current research of ASP geopolymers and suggests its possible applications in the future, which will help to provide directions for further research activities of relevant researchers.

## 2. The Preparation Process of ASP Geopolymers

### 2.1. Raw Materials

The selection of aluminosilicate precursor is an important factor affecting the properties and structure of ASP geopolymers. The commonly used aluminosilicate precursors can be divided into two categories: natural raw materials, such as metakaolin and volcanic ash, and industrial solid wastes rich in silicon and aluminum components, such as fly ash and blast furnace slag. The types and components of precursors used to synthesize ASP geopolymers are shown in Table 1.

Metakaolin has become the preferred aluminosilicate precursor because of its high activity, high aluminosilicate content, and low impurities. Previous studies have shown that the reactivity of metakaolin will be better when the Si/Al molar ratio is low [22,23], and the presence of impurities (such as quartz) in metakaolin will weaken the geopolymerization reaction and the compressive strength of ASP geopolymers [24]. In addition, it seems that the obtained geopolymer structure is denser and the compressive strength is higher with the decrease in the particle size of metakaolin [25]. However, it should be noted that metakaolin is usually obtained from natural kaolin calcined at 600~900 °C to improve its reaction activity [26]. In addition, due to the slow release of aluminum in phosphoric acid solution at room temperature, ASP geopolymers face the problem of a long setting time at room temperature (usually several days). To accelerate the reaction speed and shorten the setting time, the usual method is to increase the curing temperature (ranging from 40 °C to 100 °C). All of these operations will lead to an increase in energy consumption and cost in practical applications of ASP geopolymers.

**Table 1 materials-15-05961-t001:** Main components of commonly used aluminosilicate precursors.

Precursor Type		Main Chemical Compositions (wt.%)									Ref.
		SiO_2_	Al_2_O_3_	CaO	Fe_2_O_3_	MgO	TiO_2_	Na_2_O	P_2_O_5_	LOI	
Natural aluminosilicate precursors	Metakaolin	41–75	22–44	0.01–0.91	0.23–7.65	0.06–0.65	0.49–4.45	0.03–0.62	0.02–0.49	0.1–2.43	[26]
	Kaolinite	47.69	36.48	0.08	0.69	0.11	0.36	0.07	/	13.47	[27]
	Volcanic ash	39–45	16.9–17.7	8.1–8.7	11.4–14.3	5.41–7.65	2.70–3.42	1.7–2.9	0.63–1.01	3.35–4.81	[28]
	Volcanic ash	41.66	15.98	9.26	13.51	8.18	3.01	/	0.89	4.25	[29]
	Raw laterite	52.3	21.68	0.08	10.68	0.12	1.29	/	0.1	/	[30]
	Raw laterite	35.53	28.21	0.21	36.32	0.25	1.81	0.82	/	/	[31]
	Halloysite	46	37.8	0.07	0.72	0.13	0.07	/	/	14.9	[32]
Solid wastes	Fly ash	49.07	32.38	3.43	7.80	0.55	/	0.06	1.01	2.30	[33]
	Fly ash	53.63	21.71	10.80	7.96	1.17	0.86	1.20	/	0.33	[34]
	LCFA *	44.5	31.2	5.3	6.5	1.9	1.2	1.1	/	3.8	[35]
	HCFA *	38.1	26.5	16.5	8.5	1.2	1.7	0.6	/	6.5	
	GGBFS *	38.0	10.8	40.1	0.3	7.24	0.83	0.31	0.02	/	[36]
	EMDR *	10.36	4.279	0.064	8.739	0.09	/	0.052	0.096	/	[37]
	Mine tailings	16.2	2.6	0.4	38.9	/	0.2	/	0.3	28.1	[38]
	SFCC *	37.63	55.29	0.39	0.58	/	/	0.15	/	/	[39]

LCFA *: low-calcium fly ash. HCFA *: high-calcium fly ash. GGBFS *: ground granulated blast furnace slag. EMDR *: electrolytic manganese dioxide residue. SFCC *: spent fluid catalytic cracking catalyst.

Exploring the use of solid wastes rich in silicon and aluminum as aluminosilicate precursors will help to reduce the consumption of natural raw materials and reduce carbon emissions. Fly ash (FA) is a kind of industrial solid waste mainly composed of CaO, Al_2_O_3_, and SiO_2_ produced by coal-fired power plants. FA is usually divided into high-calcium fly ash (HCFA) and low-calcium fly ash (LCFA) according to the content of CaO. The micromorphology of fly ash particles is mostly a spherical ball shape, as shown in Figure 3a,b. Figure 3c shows the geopolymerization reaction mechanism of fly ash and phosphoric acid. Fly ash undergoes a dealumination reaction under the erosion of phosphoric acid and then geopolymerization to form amorphous products composed of –Si–O–P–, –Si–O–Al–P–, AlPO_4_, and some new crystallization products such as CaHPO_4_ and CaPO_3_(OH)·2H_2_O [33]. Figure 3d,e shows that the addition of HCFA can shorten the setting time and improve the early strength of the geopolymer. The addition of LCFA can improve the fluidity of the geopolymer but will have an adverse effect on the setting time and compressive strength [35]. Although fly ash can provide rich calcium to improve the early strength of geopolymers, long-term studies have found that calcium phosphate formed by the reaction will gradually transform into needle-like and flaky particles, which affects the long-term strength of geopolymers [34,40]. In addition, other studies found that the addition of solid wastes such as blast furnace slag and electrolytic manganese slag can accelerate the geopolymerization reaction, and the prepared geopolymer samples have an excellent water resistance and a high temperature resistance [36,37]. The disadvantages of solid wastes are that their compositions are complex and volatile, and they are rich in harmful heavy metal ions and other pollutants, such as dioxin. These problems have always been the main reasons that restrict its large-scale utilization in cementitious materials.

In addition, the choice of activator is also an aspect worth considering. In most studies, phosphoric acid is the most commonly used activator. The use of phosphoric acid as an activator can eliminate the interference of other components, which is convenient to analyse and study the reaction mechanism and microstructure of ASP geopolymers. However, the high price of phosphoric acid and the depletion of phosphate rock for the production of phosphoric acid worldwide will make the large-scale production of phosphate geopolymers unsustainable [3]. Looking for alternative activators of phosphoric acid has important research value and significance; for example, the use of waste disused phosphoric acid-based polishing liquid is a good attempt [41]. The performance of phosphate geopolymers prepared by using other waste solutions containing phosphoric acid or phosphates instead of phosphoric acid are not poor; in contrast, admixture such as metal ions contained in waste solutions can promote the geopolymerization reaction and optimize the microstructure of ASP geopolymers [41].

Ordinary Portland cement (OPC) is one of the most commonly used inorganic cementing materials, and widely used in human society. However, the production of OPC has caused considerable energy consumption and greenhouse gas emissions, which makes the sustainable development of human society difficult. The application of geopolymer materials is considered to be an effective solution to the environmental and energy problems caused by the OPC industry [42]. It has been reported that the overall energy consumption and carbon emissions during the production of ASP geopolymers are only 25% and 20% of those of the OPC industry [7,15], which makes the ASP geopolymers more environmentally friendly and sustainable than OPC in terms of gas emissions and energy consumption. Aluminosilicate industrial solid wastes (e.g., blast furnace slag, fly ash) can be used as ideal aluminosilicate precursors to synthesize high-performance ASP geopolymers [43], while they can only be used as supplementary cementing materials in the OPC system [44]. In addition, the application of ASP geopolymer materials may also form a benign ecological cycle system in the environment. The phosphorus in geopolymer materials may gradually evolve into new mineral deposits or be absorbed by plants as fertilizer under chemical and biological action [45,46]. Recycling in the ecosystem will make the ASP geopolymer materials better achieve environmental sustainability.

### 2.2. Curing Conditions

The curing temperature is an important parameter in the preparation process of ASP geopolymers and strongly affects the curing time and properties of geopolymers. AAS geopolymers usually can rapidly solidify and harden at room temperature while ASP geopolymers cannot, which is mainly related to the slow release of silicon and aluminum in aluminosilicate precursors in acidic environments. To accelerate the curing reaction of geopolymers, scholars have made many attempts, such as heating and microwaves [47]. Heating to increase the curing temperature has become the most important means to accelerate the curing reaction of ASP geopolymers because of its simple operation. In the published literature, we have noticed that some scholars have used room temperature in their studies [29,48], while most others have used elevated temperature curing. The main heating range is between 40 °C and 90 °C, such as 40 °C [23], 60 °C [17,19,41,49,50], 80 °C [51,52,53], and 90 °C [54]. There is no obvious promoting effect when the heating temperature is too low, and if the heating temperature is too high, it may lead to the cracking of the geopolymer structure or transformation into other types of materials, such as zeolite [1,55]. Celerier et al. and Zribi et al. found that an increase in curing temperature can promote the dealumination reaction of metakaolin and accelerate the release of aluminum and geopolymerization reactions, shortening the curing time from a few days at room temperature to a few hours at 70 °C [22,56]. The obtained geopolymer samples show an amorphous composite structure composed of two kinds of geopolymer networks at two curing temperatures: one is based on the –Al–O–P– unit, and the other is based on the –Si–O–T– unit (T = Si, Al and P), such as the –Si–O–Al–P– structure. In addition, with an increasing curing temperature, the content of aluminum phosphate in the geopolymer structure is higher; thus, the compressive strength is improved. In addition, it has also been reported that ASP geopolymers are prepared by multistage curing temperature process [34,55,57], such as pre-curing at 40 °C and then further curing at 80 °C. As shown in Figure 4, Lin et al. adopted a two-stage temperature curing method, which successfully avoided expansion cracking of geopolymers during high-temperature curing [55]. In addition, curing relative humidity also seems to have an effect on the properties of the ASP geopolymers. In the literature on ASP geopolymers, most researchers only pay attention to the effect of curing temperature on the preparation of geopolymers but ignore the possible effect of curing environment relative humidity on the preparation of geopolymers. Only a small number of researchers deliberately set the curing relative humidity to ≥90% in their studies [40,57,58], but did not explain the reason and the resulting impact on material properties and structure. In the study of Dong et al., we found that it seems that the relative humidity of the curing environment does not have an insignificant effect on geopolymers as many researchers believe. Dong et al. solidified two groups of samples at 3% and 98% humidity with the same other conditions. After a comparative study, it was found that a high humidity was beneficial to the formation of AlPO_4_ in the geopolymer structure; thus, the samples with a higher compressive strength were obtained [59]. Therefore, the curing relative humidity is also a noteworthy aspect in future ASP geopolymer research.

### 2.3. Geopolymerization Mechanism

At present, there is no exact definition and unified consensus on the reaction mechanism and microstructure of ASP geopolymers, and many research results even contradict each other. In early studies, it was generally believed that the geopolymerization mechanism of geopolymers is an adhesion reaction between the aluminum unit dissolved by aluminosilicate precursors in phosphoric acid solution and phosphorus–oxygen tetrahedron. Alternatively, it is considered that the [SiO_4_]^4−^ unit is partially replaced by the [PO_4_]^3−^ unit after the Si–O–Al bond is broken in the acid medium, resulting in an amorphous geopolymer network structure based on Al–O–P, Si–O–P and Si–O–P–O–Al [16,17,48].

In fact, with the deepening of research and the progress of characterization methods, researchers have gradually found that the structure of ASP geopolymers is actually more inclined to be a kind of amorphous and crystalline composite structure; amorphous structures such as Al–O–P, Si–O–Al–O–P and Si–O–P–O–Al are dominant, while crystalline structures such as AlPO_4_ are dispersed in the amorphous matrix to form the geopolymer structure [25,50,60,61]. The formation of the crystal phase is affected by many factors, such as the phosphoric acid concentration, the curing temperature, and the particle size of the aluminosilicate precursor [25]. Zribi et al. found that the prepared geopolymers show an amorphous composite structure: one is the geopolymer network based on Al–O–P, and the other is the geopolymer network based on Si–O–Al–O–P under the condition of a lack of acid (P/Al = 0.5). In addition, it shows a completely different structure, which is mainly composed of an aluminum phosphate hydrate crystal phase and some amorphous geopolymer network when the acid is excessive (P/Al = 2), indicating that excess phosphoric acid can induce the crystallization of the geopolymer structure [62].

At present, with regard to research on the reaction mechanism of ASP geopolymers, scholars have generally divided the geopolymerization process of ASP geopolymers into three or even four steps with depolymerization (the dealumination process) and condensation as the core [20,25,49]. The mainstream geopolymerization schematic diagram is shown in Figure 5, and the mainstream geopolymerization process is as follows:The first is the dealumination process of the aluminosilicate precursor. The silicon–oxygen tetrahedron and aluminum–oxygen tetrahedral structure of aluminosilicate precursors depolymerize under the action of phosphoric acid, mainly including the breaking of Al–O–Al bonds and Si–O–Al bonds and the formation of free Al^3+^ and –Si–O– units.Then, there is the polycondensation of PO_4_^3−^, Al^3+^, and –Si–O– units and the formation of crystalline phases such as AlPO_4_.The final geopolymerization process is based on the previous polycondensation reaction, and the units are further condensed to form larger geopolymerization chains and form different three-dimensional geopolymer network structures. At the same time, the crystal phase, such as AlPO_4_, is dispersed in the amorphous geopolymer network structure and finally forms a kind of amorphous and crystalline composite geopolymer structure.

## 3. The Properties of ASP Geopolymers

### 3.1. Mechanical Properties

At present, research on the mechanical properties of ASP geopolymers is almost focused on mechanical strength, and few scholars pay attention to other aspects, such as impact toughness, ductility, and bond properties. ASP geopolymers show higher compressive strengths than Portland cement and AAS geopolymer [17,21]. Perera et al. found that the maximum compressive strength of ASP geopolymers obtained is 146 MPa, which is much higher than the 72 MPa of AAS geopolymers obtained under the same curing conditions [17]. This shows that ASP geopolymers have great application potential.

The data on the compressive strength of ASP geopolymers in the available literature fluctuate widely. The highest compressive strength recorded in the literature is 146 MPa [17], and the lowest compressive strength appears in the ASP geopolymer foam, which is only 0.64 MPa [63]. In fact, the compressive strength of ASP geopolymers recorded in the literature is affected by many factors, such as the composition, purity, particle size, and activation conditions of aluminosilicate precursors. In addition, the type and concentration of the activator, the curing temperature, and the humidity are also important influencing factors, and the effects of different factors on the compressive strength of ASP geopolymers are shown in Table 2.

Table 3 establishes the compressive strength database of ASP geopolymers, recording the compressive strength obtained by the synthetic process selected in different studies. There are significant differences in the compressive strength of ASP geopolymers prepared by different aluminosilicate precursors. At present, metakaolin is the most commonly used aluminosilicate precursor, but some scholars have made other attempts, such as using electrolytic manganese slag to obtain geopolymers with compressive strengths up to 70.8 MPa [37], volcanic ash 81.3 MPa [67], fly ash 76 MPa [34], laterite 82.6 MPa [31], halloysite 26 MPa [32], and so on. The concentration of phosphoric acid also has an important effect on the compressive strength of the geopolymer. The excitation effect of raw materials is not good when the amount of phosphoric acid is insufficient, while the excessive [PO_4_]^3−^ unit will cause a charge imbalance when the amount of phosphoric acid is too high, resulting in a looser microstructure and decreased mechanical properties and water resistance of ASP geopolymers [56]. In addition, too much phosphoric acid will also lead to an increase in the volume shrinkage, cracking rate, porosity, and pore size of the geopolymer matrix [68]. At present, scholars generally believe that the compressive strength will first increase and then decrease with increasing concentrations of phosphoric acid, but the understanding of the optimal phosphoric acid concentration or Al/P ratio is still a problem that cannot be unified. Dong and Hervé et al. found that the best concentration of phosphoric acid obtained by both of them is 10 mol/L [61,65]. There are also studies on the effect of the P/Al molar ratio on the compressive strength of geopolymers. The results showed that the best mechanical property can be obtained when the P/Al molar ratio is one [23,62,69]. However, some studies have found that the optimal P/Al molar ratio is less than one [55,59]. Due to the difference in raw material composition and activity, the same experimental results cannot be obtained even if metakaolin is used as the same precursor, which makes it difficult to establish standard synthesis methods and performance indicators of ASP geopolymers.

### 3.2. Heat and Fire Resistance

Previous studies have shown that ASP geopolymers have excellent heat resistances and good thermal stabilities, and the melting phenomenon of the geopolymer sample is not observed even at temperatures as high as 1550 °C [18,22,71]. This kind of heat resistance is not available in AAS geopolymer materials and ordinary Portland cement materials. AAS geopolymer materials will have some problems, such as “panalkali” and fiber deterioration at high temperature, due to the existence of alkaline melting cations, while Portland cement will peel off or even burst inside at high temperature. At present, research on the heat resistance of ASP geopolymers is mainly focused on high-temperature thermal stability and crystal phase transformation. The results of Liu et al. showed that ASP geopolymers have excellent thermal stabilities at high temperatures [18]. As shown in Figure 6, the geopolymer crystallized to form quartz and sphalerite when heated to 900 °C, and when heated to 1150 °C, these two crystals transformed into cristobalite and aluminum phosphate, respectively. Finally, the cristobalite and aluminum phosphate phases transformed into needle-like structures when heated to 1550 °C, but no signs of sample melting were observed. In addition, the amorphous geopolymer network formed by the reaction of metakaolin with phosphoric acid will be transformed into crystalline AlPO_4_ at high temperature, and the silicon-containing phases will be rearranged to form new AlSi_2_(PO_4_)_3_ compounds. The reaction formula of the specific evolution process is shown in Equation (3).
(3)$$\begin{array}{l} {\left(–\mathrm{Si}–O–\mathrm{Al}–O–P–O–\right)}_{n}\left(\mathrm{Geopolymer}\right)\\ \overset{900\;{}^{\circ}C}{\to }{\mathrm{SiO}}_{2}\left(\mathrm{Quartz}\right){+\mathrm{AlPO}}_{4}\left(\mathrm{Berlinite}\right)\overset{1150\;{}^{\circ}C}{\to }{\mathrm{SiO}}_{2}\left(\mathrm{Cristobalite}\right){+\mathrm{AlPO}}_{4}\left(\mathrm{Aluminum}\;\mathrm{phosphate}\right){+\mathrm{Al}}_{6}{\mathrm{Si}}_{2}O_{13}\left(\mathrm{Mullite}\right) \end{array}$$

In addition, the concentration of phosphoric acid will also affect the formation of new phases of geopolymer at high temperatures [55,60,72]. Douiri et al. found that the samples with a high Al/P molar ratio (Al/P = 4) do not have a complete amorphous geopolymer network structure, while the samples with a low Al/P molar ratio (Al/P = 1) usually contain a complete geopolymer network structure, which ensures structural stability when heated at a high temperature. As shown in Figure 7, by using XRD and other characterization methods, Sellami et al. found that the zeolite phase in the spectrum was crystallized at 180 °C and transformed into the phosphor-cristobalite phase at 300 °C. Moreover, a new tridymite SiO_2_ phase was observed at 700 °C and kept at 1400 °C. At approximately 1100 °C, the number of phosphor-cristobalite and tridymite crystals reaches the maximum, which is similar to the behavior observed in Liu et al. [18,73]. However, Bewa et al. observed that a new phospho-tridymite or phospho-cristobalite phase can be formed at a lower temperature of 200 °C [74].

In addition, previous studies have shown that ASP geopolymers also have good high-temperature thermal stability and low thermal conductivity and can withstand the fire resistance test of a 1100 °C flame [71]. The amorphous aluminosilicate phosphate phase in the ASP geopolymers will be transformed into cristobalite and aluminophosphate crystalline phases at high temperatures [41].

### 3.3. Dielectric Property

ASP geopolymer materials have excellent dielectric properties and are suitable for use as potential new insulator materials [19,53]. Water molecules, hydroxyl ions, and free metal ions such as sodium ions are the main components that determine the dielectric loss at different radio frequencies in AAS geopolymer materials, so it is difficult to reduce the dielectric loss caused by ion transfer. Different from AAS geopolymers, ASP geopolymers are mainly composed of silicon–oxygen tetrahedron, aluminum–oxygen tetrahedron, and phosphorus–oxygen tetrahedron. In the whole geopolymer network structure, the charge between aluminum–oxygen tetrahedron with a negative univalent and phosphorus–oxygen tetrahedron with a positive univalent can be balanced with each other, making the whole system electrically neutral [2]. Moreover, the dielectric loss caused by ion transfer is very low due to the lack of metal cations (such as Na^+^ or K^+^) in ASP geopolymers, so the order of magnitude of dielectric loss can be reduced from 10^−2^ to 10^−3^ by heat treatment, which has potential application value in packaging materials [53]. The study of Sellami et al. showed that the dielectric constant of ASP geopolymers increases with increasing temperature, which is due to the enhancement of ion conduction observed in the low frequency range. However, the electrical conductivity of the geopolymer material is no more than 10^−7^ s cm^−1^ for the highest temperature used in the literature (725 °C), so it is a good insulator [73]. The dielectric loss caused by ion transfer can be ignored by reducing or eliminating free water, and the appropriate heat treatment temperature can ensure that the ASP geopolymers have the best dielectric property [48]. In addition, as shown in Figure 8, Douiri et al. confirmed that the dielectric constant increases with a decreasing Si/P molar ratio, i.e., with an increasing phosphoric acid content. The increase in the permittivity and conductivity of ASP geopolymers may be due to the existence of an additional charge center caused by an additional proton provided by the phosphoric acid molecule. ASP geopolymers can be regarded as insulator materials with good dielectric properties regardless of whether the Si/P molar ratio is 1.25, 1.5, or 1.75 [19].

### 3.4. Durability

The water resistance of ASP geopolymers seems to be unsatisfactory. Bewa et al. found that the compressive strength of the sample decreased by 54% after soaking in water for 28 days [74], while the compressive strength of ASP geopolymers prepared with natural laterite lost more than 60% after soaking in water for 24 h by Mimboe et al. [30]. By means of FIIR and other characterization methods, Bewa et al. attributed the problem to structures such as –Si–O–P– bonds in the structure that are easy to dissolve in water to form silanol (Si–OH) and P–OH groups, resulting in the deterioration of the mechanical properties of the geopolymer samples.

The ASP geopolymer samples with fine water resistance usually contain rich crystalline phases, such as cristobalite or bainitic AlPO_4_, while the non-water-resistant samples contain a large number of amorphous phases. In addition, NMR results showed that a higher orderliness in the geopolymer network structure is beneficial to improve the water resistance, and a higher orderliness usually represents a more complete crystallization [75].

The study of the acid and alkali resistance of ASP geopolymers will help to expand its application range in the future, such as in marine materials. Previous studies have shown that ASP geopolymers have poor acid and alkali resistance. Table 4 shows that the ASP geopolymers will react in an acidic or alkaline environment, resulting in poor mechanical properties or even complete decomposition of the sample matrix structure [37]. The current research mostly evaluates the durability of ASP geopolymers from the perspective of surface structure changes such as cracks and mechanical property losses, which lack evidence for microscopic characterization. In addition, the current durability evaluation cycle is too short, which cannot reflect the long-term durability of ASP geopolymers in acidic and alkaline environments. As the performance to resist the deterioration of environmental media, the durability of ASP geopolymer materials plays a crucial role in the service life and safety of materials. The resistance to carbonation/weathering, sulfate corrosion, chloride ion corrosion, freeze–thaw cycles, and water of the matrix will be the focus of the durability study of ASP geopolymer materials in the future [11,76,77]. Establishing scientific and standardized durability test methods and performance evaluation standard systems will help more researchers to carry out relevant work and the industrial application of materials.

## 4. The Modification of ASP Geopolymers

### 4.1. Modified with Admixture

Based on the principle of chemically bonded phosphate, many admixtures rich in calcium, magnesium, and iron have been used for the modification of ASP geopolymers to improve their workability and mechanical properties [34,58,67]. The dissolution rate of divalent metal cations in an acidic environment is usually higher than that of trivalent metal cations [78]; therefore, Ca^2+^, Mg^2+^, and Fe^2+^ in the admixture will quickly dissolve out before Al^3+^ is dissolved and reacts with phosphoric acid to form new phosphate compounds at the beginning of the geopolymerization reaction of ASP geopolymers. Wang and Guo et al. found that a large amount of calcium in fly ash will quickly dissolve out and react with phosphoric acid to form calcium phosphate in the early stage of the geopolymerization reaction, and calcium phosphate will combine with water, which makes the sample coagulate rapidly and shortens the setting time [35,40]. The geopolymer network structure induced by dealumination of aluminosilicate precursor has not been formed and cannot form an effective compressive strength in the early stage. However, the addition of calcium will form calcium phosphate nucleation sites through these nucleation sites to combine the loose structure and tissue, thus forming samples with a certain early compressive strength. It can be seen that different new mineral phases, such as brushite and mullite, have been formed from Figure 9a, and it is obvious that the mapping regions of Ca and P elements are distributed synchronously from Figure 9b, which is strong evidence that calcium reacts with phosphate to form a new phase. According to the viewpoint of Wang and Guo et al. [35,79], the possible reaction of calcium in ASP geopolymers can be roughly expressed by Equations (4)–(6) in Table 5. Therefore, it will form structures such as calcium phosphate, β-tricalcium phosphate, and brushite according to different degrees of reaction. The current problem is that too much calcium will compete with silicon–aluminum to react with phosphoric acid, which is not conducive to the formation of a geopolymer network. At the same time, a large number of calcium phosphate phases will gradually transform into needle-like and flaky structures in the later stage. These factors will weaken the compactness of the geopolymer later structure and are not conducive to the development of compressive strength. In addition, the formed calcium phosphate phase is unstable and will decompose and transform at a high temperature [35], as shown in Equations (7) and (8) in Table 5.

The role of magnesium and iron in ASP geopolymers is similar to that of calcium. Previous studies have shown that the addition of magnesia will produce struvite-like products in the structure of ASP geopolymers [3]. The specific reaction process can be seen in Equation (9) in Table 5. The addition of magnesia to ASP geopolymers can form a new phase, which can effectively shorten the setting time and improve the early compressive strength. In addition, the acid-base reaction between magnesium and aluminum phosphate can form a new magnesium phosphate phase and an amorphous magnesium aluminum phosphate phase (Al_2_O_3_·3MgO·2P_2_O_5_) [58]. The general reaction is shown in Equations (9) and (10) in Table 5. When iron aluminosilicate is mixed with phosphoric acid activator, the following two steps should occur: (1) dealumination and deferritization by the H^+^ ions from acid and (2) then free aluminum ions and iron ions react with phosphate (PO_4_^3−^) to form an amorphous iron aluminum phosphate phase [30]. Han et al. found that the reaction between magnetite (Fe_3_O_4_) and phosphoric acid forms a new amorphous gel phase, which may be composed of ferric hydrogen phosphate [80]. The related reaction is given by Equations (11) and (12) in Table 5 [37]. In addition, Bewa et al. found that the hematite phase (Fe_2_O_3_) can also participate in polycondensation and form a new polyphosphate siloxane (–S–O–Si–O–P–O–Si–O–Fe–O–) chain in the process of geopolymerization. The existence of hematite (Fe_2_O_3_) will form more nucleation sites, which is conducive to triggering geopolymerization, and these changes will have a positive impact on the compressive strength of geopolymer samples [24].

### 4.2. Modified with Fiber

Fiber-reinforced composites are composed of fibers, a matrix, and their interfaces. They are considered a new field of advanced composites and are widely used in engineering and manufacturing [66]. It is a common method to modify cementitious materials with different fibers, such as glass fibers and carbon fibers, which have been widely used in Portland cement-based cementitious materials and alkali-activated geopolymer materials [81]. Previous studies have shown that different fibers can also strengthen the properties and improve the microstructure of ASP geopolymers through good adhesion and different strengthening mechanisms (such as fiber bridging mechanisms and fiber drawing strengthening mechanisms). Yang et al. found that the compressive strength of geopolymer samples could be increased from 50 MPa to 110 MPa when the content of polyimide fiber was 1.5 wt%, which increased by 120% and the flexural strength increased by 283% [66]. There is good adhesion between the polyimide fiber and the geopolymer matrix, and the mechanical properties of the geopolymer materials can be significantly enhanced by the fiber bridging mechanism. Figure 10a shows the strengthening mechanism of polyimide fiber-modified ASP geopolymers. Small cracks will gradually appear in the structure when the unreinforced matrix is subjected to stress, and the growth of the crack directly affects the flexural strength of the matrix. The fiber can be used as a bridge to span the microcrack caused by stress, thus effectively preventing the growth of cracks. After that, Yu et al. incorporated multilayer SiO_2_ fibers into ASP geopolymers and found that the samples containing 17 vol% SiO_2_ fibers had the best mechanical properties [82]. The existence of SiO_2_ fibers enhanced the bonding ability of the matrix interface, which could optimize the tensile/fracture behavior of the geopolymers and increase the toughness. A recent study reported that there is good adhesion between the mullite fiber and the ASP geopolymer matrix (Figure 10b), which promotes fiber drawing strengthening and crack deflection mechanisms (Figure 10c,d) and has a significant fiber strengthening effect [83]. The best compressive/flexural strength can be obtained by adding 10% mullite fiber, while the addition of 20% mullite fiber has the strongest inhibitory effect on the shrinkage of geopolymer samples calcined at a high temperature.

Fiber-modified ASP geopolymers will be an important research field in future research. However, it should be noted that appropriate fibers should be selected according to the application direction of actual composites. For example, glass fiber is not suitable for light geopolymer materials, and carbon fiber cannot be used for composites requiring thermal insulation and electrical insulation.

In addition, the modification of ASP geopolymers by nanomaterials such as graphene is also an interesting research aspect. Graphenes have a very large specific surface area, a high strength, and toughness, and have a good binding ability with inorganic cementitious materials [84]. Previous studies have shown that the toughness and electrical properties of alkali-activated geopolymer materials have been greatly enhanced by the modification of graphene [85]. The structure and formation mechanism of ASP geopolymers are similar to those of alkali-activated geopolymer. Therefore, graphene-modified ASP geopolymers will have great research and application potential.

## 5. Applied Research on ASP Geopolymers

### 5.1. Porous Foam Materials

At present, the preparation of ASP geopolymer porous foam materials by adding foaming agents is a sufficient direction in the application research of ASP geopolymers. The ASP geopolymer porous foam materials not only have heat and fire resistance but also have the characteristics of light weight and heat insulation, making them a potential new kind of lightweight thermal insulation and fireproof wall material [18]. The foaming agents used now include aluminum powder [51,86], iron powder [86], surfactant [63], hydrogen peroxide [71], limestone [54,87], and so on. The basic principles for the preparation of ASP geopolymer foam materials are mainly divided into two categories. One is the use of foaming agents such as aluminum powder, iron powder, or limestone to react with water or acid to produce gas, which produces a uniform and rich bubble structure in the process of mixing the geopolymer paste. The other is to use the foaming agent based on surfactant, which is directly mixed with the slurry. Figure 11 shows the foaming principle of H_2_O_2_ and surfactants. Foam materials with different porosities can be obtained by controlling the content. Table 6 shows the performance parameters of ASP geopolymer porous foam materials prepared with different foaming agents. Figure 12 shows the morphology of geopolymer foams prepared by different foaming agents. It can be seen that the pore structure of the geopolymer made of limestone as the foaming agent is irregular, while that of other foaming agents is mostly round and uniformly distributed.

### 5.2. Heavy Metal Solidification/Radioactive Nuclear Waste Management

Previous studies have shown that members of the phosphorate-based geopolymer family, such as magnesium phosphate cement (MPC) and aluminosilicate phosphate (ASP) geopolymers, have excellent adsorption and solidification abilities for heavy metals and radioactive nuclear wastes [88]. Phosphorate-based geopolymers can firmly block heavy metal ions in the cavity of their unique three-dimensional network structure, which can effectively reduce the leaching of heavy metal ions or radioactive elements. Pu and Njimou et al. found that ASP geopolymers have a good stabilizing effect on Pb^2+^ ions, and their performance is better than that of AAS geopolymers and ordinary Portland cement, especially in acidic environments [89,90]. The solidification and stabilization mechanisms of ASP geopolymers to Pb^2+^ ions mainly include chemical precipitation, physical adsorption, and encapsulation. Pb^2+^ ions are mainly stabilized in the form of lead-stabilized compounds of Pb_3_(PO_4_)_2_ and PbHPO_4_ in the geopolymer matrix. ASP geopolymers can be used as low-cost and efficient adsorbents after treatment. Another study proved that heavy metal pollutants (such as Cd, Zn, or Cr) can be well fixed in apatite minerals formed by the reaction of calcium and phosphoric acid [88]. Tome et al. found that ASP geopolymers also have a certain adsorption effect on anionic (Eriochrome Black T/EBT) and cationic (methylene blue/MB) compounds in sewage. Adsorption is pH-dependent, which means that electrostatic interactions are the main driving mechanism of adsorption [52]. In addition, the attempt to cure tributyl radioactive phosphate (TBP, a waste organic solvent containing a variety of radionuclides such as uranium and plutonium) with ASP geopolymers has also demonstrated its good curing efficiency for radioactive nuclear waste [91].

### 5.3. Possible Applications in the Future

Referring to the literature related to ASP geopolymers, there are few reports on their application research. ASP geopolymers have many excellent properties, such as high compressive strengths (more than 140 MPa) [17], excellent heat resistances [18], and good dielectric properties [19,53]. However, there are also some unavoidable problems in ASP geopolymers, such as a high preparation cost, a long condensation hardening period at room temperature, a poor water and corrosion resistance, and so on. We can predict the possible application direction and possible products of ASP geopolymers in the future according to the advantages and disadvantages.

The excellent mechanical properties and heat and fire resistance of ASP geopolymers make it possible for them to be used as building materials. Studies of ASP geopolymer cements or ASP geopolymer concretes will promote their application in building materials. However, it should be noted that the problem of a slow setting time at room temperature needs to be well solved when ASP geopolymers are used in building materials, and the long hardening time is not conducive to the development and progress of construction. In addition, we can change the angle and use it in an environment that can provide a high temperature. For example, it may be a good attempt to use ASP geopolymers as oil well cement. The high temperature of 60~80 °C under normal oil wells can provide conditions for ASP geopolymers to accelerate the curing reaction. ASP geopolymers also have the potential to be used as new lightweight thermal insulation wall materials [92]. Previous studies have shown that light ASP geopolymer foams made by foaming agents have excellent heat and fire resistance, so they are very suitable for use as lightweight thermal insulation materials in buildings [51,86]. In addition, ASP geopolymers can also be used as a new kind of coating. The new coating prepared by ASP geopolymers has a good hardness, wear resistance, spectral selectivity, and strong adhesion to the coating substrate and can be applied to the surface anti-corrosion and protection of many substrates [93,94]. In addition, it could also make full use of the excellent dielectric properties and heat resistance of ASP geopolymers to explore their applications in high-temperature electronic packaging or aerospace and other fields guided by market demand [19].

### 5.4. Discussion and Recommendations

Although ASP geopolymers are reliable in performance and characteristics, most of the current studies are still at the laboratory stage. Optimizing the synthesis process, developing more economical raw materials, and standardizing test methods and technical indicators will promote the industrial application of ASP geopolymers. Current problems and recommendations in the development of ASP geopolymers are as follows:The use of natural raw materials is not conducive to the sustainable development of ASP geopolymers, and the exploration of solid waste utilization should be strengthened. It is necessary to focus on solving the problem of unstable performance caused by the fluctuation of solid wastes composition and paying attention to the leaching of harmful heavy metals in solid wastes.The gradual shortage of phosphate rock resources and the high price of phosphoric acid will be the key factors limiting the wide application of ASP geopolymers in the future. It will be of great significance to actively expand the range and types of acid activators, such as the application of waste liquid containing phosphoric acid.The durability enhancement and mechanism of ASP geopolymers should be further studied owing to their poor durability.The long-term performance of ASP geopolymers should also be evaluated, with emphasis on improving the acid and alkali resistance. In addition, the environmental impact assessment of such materials is also crucial. To date, there is a lack of relevant research, which will produce some risks in the application of this material.Standardizing the test methods, technical indicators, and normalizing the preparation process of ASP geopolymers will help to promote the standardized and industrialization application of ASP geopolymers.ASP geopolymers are better than AAS geopolymers in mechanical properties, heat resistance, and dielectric properties. However, the cost of ASP geopolymers is significantly higher than that of AAS geopolymers. Therefore, the goal of AAS geopolymers is to replace traditional Portland cement on a large scale as much as possible in the future, while ASP geopolymer materials are expected to be used in some high-valued fields, such as coatings, fire-resistance, and thermal insulation materials.

## 6. Conclusions

This paper reviews the latest research progress of aluminosilicate phosphate (ASP) geopolymers in terms of their synthesis processes, performances, modifications, and applications. The current investigation indicates that ASP geopolymers have the characteristics of a low-carbon synthesis process, high mechanical properties, a strong heat resistance, and excellent dielectric properties. These excellent properties make ASP geopolymers green cementitious materials with broad application prospects in the fields of new building materials, coating materials, insulating materials, and heavy metal curing. Effectively reducing the preparation cost, enriching the variety of raw materials, and improving the durability will be the focus of future research on ASP geopolymers. Standardizing the test methods, technical indicators, and normalizing the preparation process, will help to promote the industrialization application of ASP geopolymers.

## Figures and Tables

**Figure 1 materials-15-05961-f001:**
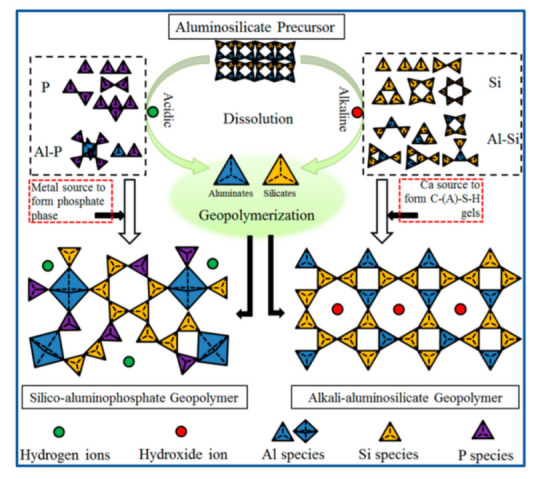
Reaction process of aluminosilicate precursors forming geopolymer under alkaline and acidic conditions [3].

**Figure 2 materials-15-05961-f002:**
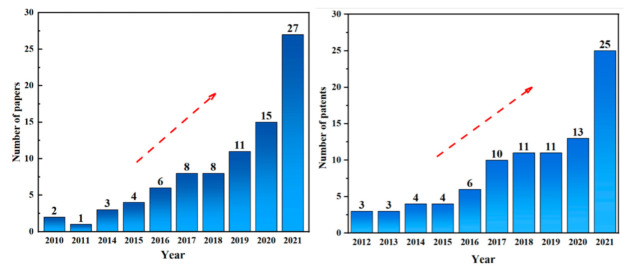
The number of papers and patents published on ASP geopolymers in recent years.

**Figure 3 materials-15-05961-f003:**
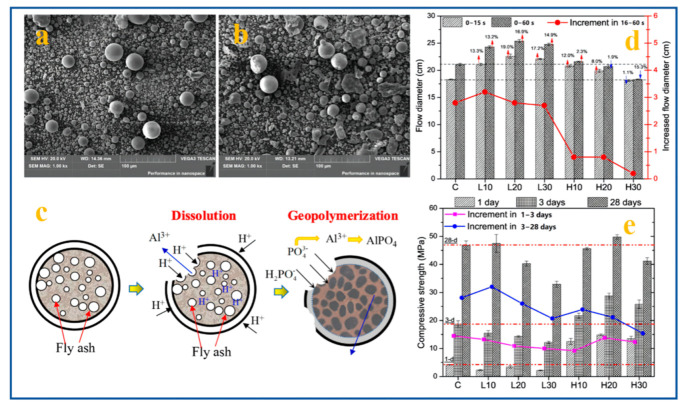
(**a**) SEM image of LCFA, (**b**) SEM image of HCFA, (**c**) geopolymerization mechanism for FA-ASP geopolymers, (**d**) fluidity of the ASP geopolymers with different types and contents of FA, and (**e**) compressive strength results of the FA-ASP geopolymers with different types and contents of FA [33,35].

**Figure 4 materials-15-05961-f004:**
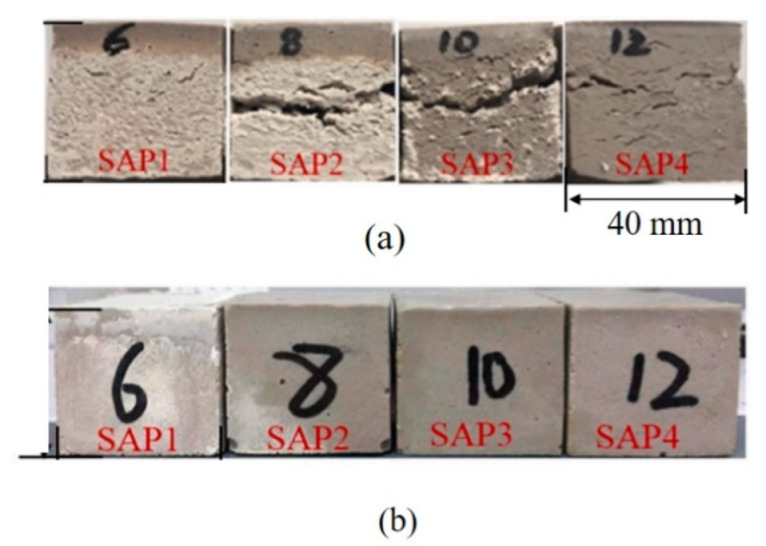
Appearance of ASP geopolymers cured under different curing conditions ((**a**) Cured at 60 °C (**b**) Cured by the two-stage curing method (precuring at 40 °C, then cured at 60/80 °C)) [55].

**Figure 5 materials-15-05961-f005:**
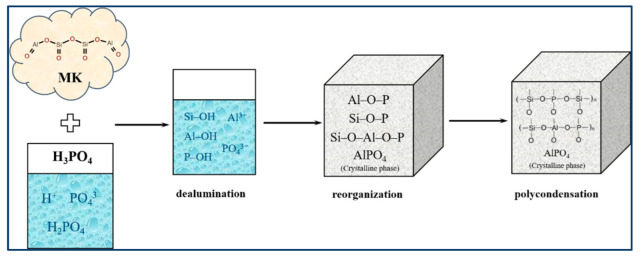
Geopolymerization mechanism of ASP geopolymers (metakaolin is used as aluminosilicate precursor).

**Figure 6 materials-15-05961-f006:**
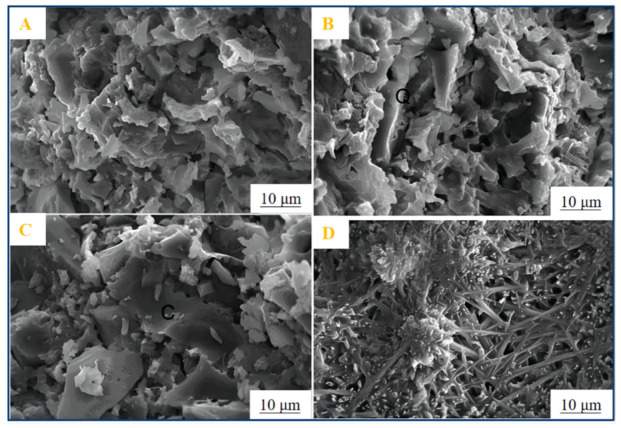
SEM images of geopolymers heated for 1 h at various temperatures (**A**)—150 °C, (**B**)—1050 °C, (**C**)—1450 °C, (**D**)—1550 °C [18].

**Figure 7 materials-15-05961-f007:**
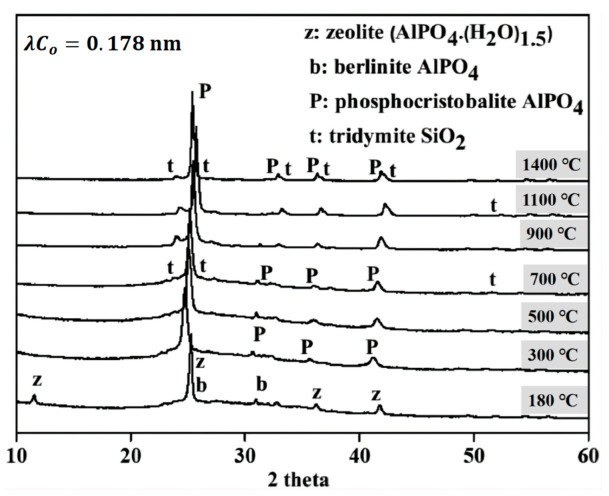
Phase evolution of geopolymer at elevated temperature [73].

**Figure 8 materials-15-05961-f008:**
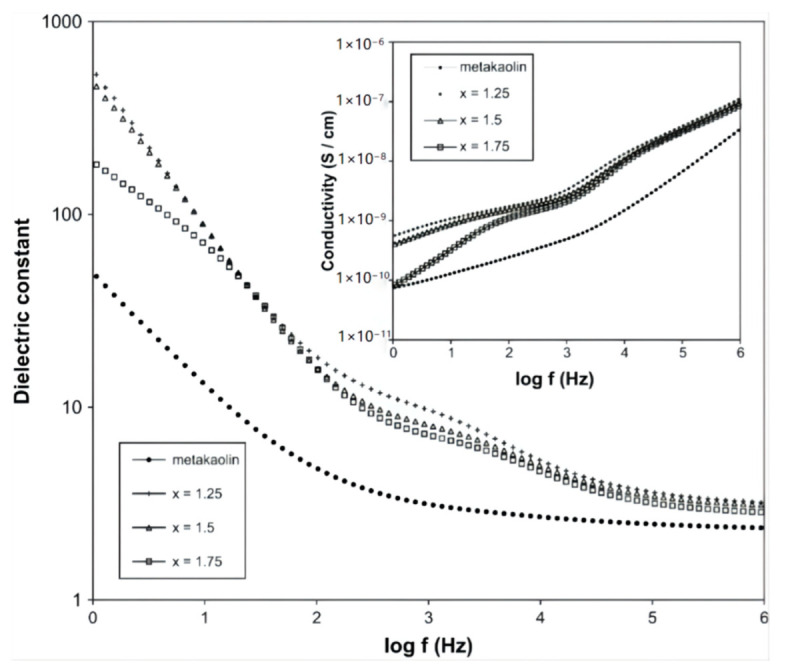
Dielectric permittivity and the inset show the conductivity of ASP geopolymers (x represents the Si/P molar ratio) [19].

**Figure 9 materials-15-05961-f009:**
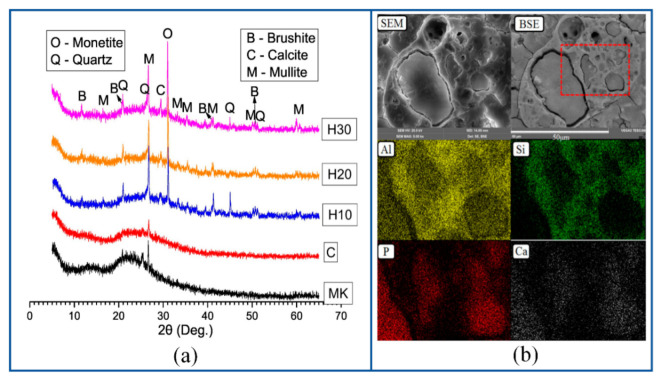
(**a**) XRD patterns of the HCFA-ASP geopolymers and (**b**) SEM/BSE images and elemental mapping results of the H30 geopolymer [35].

**Figure 10 materials-15-05961-f010:**
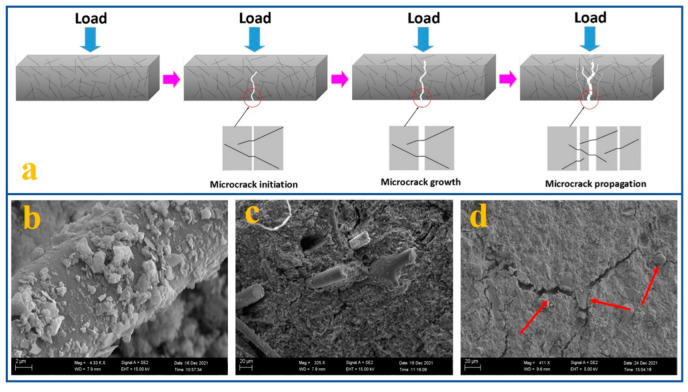
(**a**) Reinforcement mechanism of polyimide fiber-modified ASP geopolymers; SEM images of the interface between the mullite fiber and geopolymer matrix: (**b**) geopolymer grains on the mullite fibers, (**c**) fiber pullout, and (**d**) crack deflection [66,83].

**Figure 11 materials-15-05961-f011:**
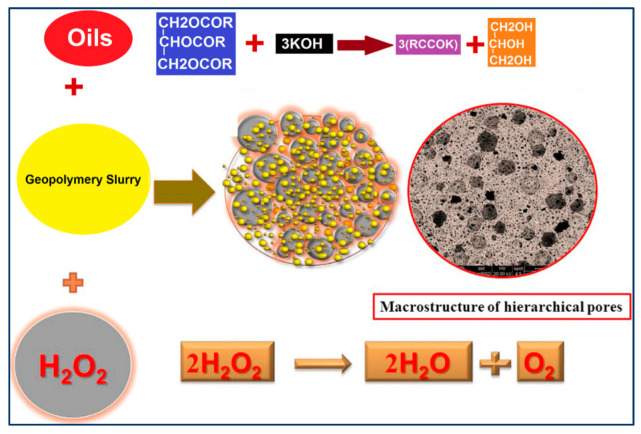
Formation of foamed geopolymers by H_2_O_2_ and surfactant [6].

**Figure 12 materials-15-05961-f012:**
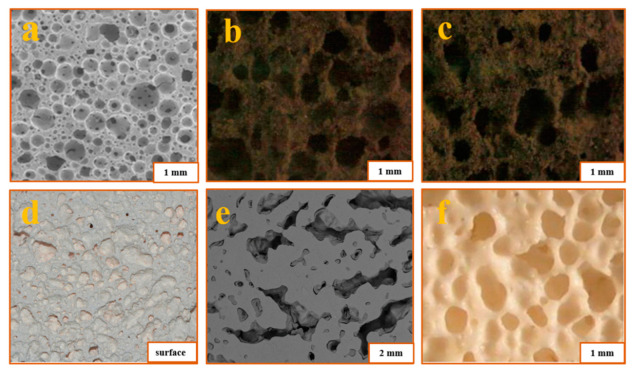
(**a**) SEM image of 15.7 mass% surfactant samples. (**b**) Image of 10 mass% Fe samples. (**c**) Image of 10 mass% Al samples. (**d**) Surface of H_2_O_2_ samples. (**e**) BSE image of limestone samples. (**f**) 0.20 mass% Al samples calcined at 1250 °C for 1 h [51,63,71,86,87].

**Table 2 materials-15-05961-t002:** Factors affecting the compressive strength of ASP geopolymers.

Influence Factors	Results	Ref.
Impurities in aluminosilicate precursors	Impurities will weaken the compressive strength	[24]
Fineness of aluminosilicate precursor	The finer the particles are, the higher the compressive strength	[25]
Activation mode of aluminosilicate precursors	Mechanical activation is better than thermal activation	[64]
Phosphoric acid concentration	The compressive strength first increased and then decreased with increasing phosphoric acid concentration	[61,65]
Curing temperature	Properly increasing the curing temperature can improve the compressive strength	[56]
Curing relative humidity	The compressive strength of samples curing at 98% humidity is better than 3% humidity	[59]
Add fibers	Appropriate fiber content can effectively improve the compressive strength	[66]

**Table 3 materials-15-05961-t003:** Compressive strength database of ASP geopolymers obtained by different synthetic processes.

Aluminosilicate Precursors	Activator	Molar Ratio/Phosphoric Acid Concentration	Liquid Solid Ratio	Curing System	Age	Compressive Strength (MPa)	Ref.
Metakaolin	H_3_PO_4_	Si/Al = 0.96 P/Al = 0.52–0.84	1.0	Precuring at 40 °C for 24 h; then curing at 60/80 °C for 24 h, respectively	3 days	60 °C 123.4 80 °C 96.8	[55]
Metakaolin	H_3_PO_4_	4–14 mol/L	0.8	Curing at room temperature for 24 h; then curing at 60 °C for 24 h	28 days	93.8	[61]
Kaolin	H_3_PO_4_	10–14 mol/L	0.9	Precuring at 40 °C for 48 h; then curing at 80 °C for 48 h	7 days 28 days	7 days 32 28 days 45	[27]
Metakaolin	H_3_PO_4_	/	1	Partial cured at room temperature; Partial cured at 60°C	15 days	RT 20.7 60 °C 29.9	[56]
Tunisian clay	H_3_PO_4_	Si/P = 2.75	/	Curing at 60 °C for 24 h	28 days	34	[25]
Metakaolin	H_3_PO_4_	10	0.8	Curing at 60 °C for 24 h	28 days	93.8	[21]
Metakaolin and MgO	Al(H_2_PO_4_)_3_	/	0.5	Curing at 25 °C and 90% relative humidity	1 days	8.3	[58]
Electrolytic manganese slag	H_3_PO_4_	/	1.0	Curing at room temperature for 24 h; Curing at 80 °C for 2 days	28 days	RT 49.8 80 °C 70.8	[37]
Volcanic ash	H_3_PO_4_	/	0.4–0.52	Curing at room temperature	28 days	81.3	[67]
Volcanic ash	H_3_PO_4_	P_2_O_5_/H_2_O = 0.12	0.45	Curing at room temperature	28 days	50.9	[28]
Fly ash	H_3_PO_4_	P/Al = 1 Si/Al = 0.91	1.04	Precuring at 40 °C and 90% relative humidity for 6 days; then curing at 80 °C for 24 h	100 days	76	[34]
Fly ash	H_3_PO_4_	Ca/P = 2.34	0.35	Curing at room temperature	28 days	50	[35]
Laterite	H_3_PO_4_	/	0.8	Precuring at 40 °C for 7 days; then curing at 65 °C for 2 days	9 days	38	[30]
Laterite	H_3_PO_4_	10 mol/L	0.8	Curing at room temperature	28 days	82.6	[31]
Metakaolin	Disused polishing liquid	/	1	Curing at 60 °C for 7 days	7 days	63–67	[41]
Halloysite	H_3_PO_4_	/	1.3	Precuring at 50 °C for 48 h; then curing at 80 °C for 48 h	28 days	25	[32]
Metakaolin	H_3_PO_4_	Si/Al = 1; P/Al = 1	/	Curing at 60 °C for 24 h	/	146	[17]
Metakaolin	Al(H_2_PO_4_)_3_	Al/P = 1/3	0.8	Curing at room temperature	28 days	32	[70]
Al_2_O_3_–2SiO_2_ powders	H_3_PO_4_	SiO_2_/Al_2_O_3_ = 1 H_3_PO_4_/Al_2_O_3_ = 1	/	Curing at 80 °C for 24 h	33 days	89.3	[53]
Metakaolin	H_3_PO_4_	P/Al = 0.6	0.3	Curing at 60 °C and 98% relative humidity for 7 days	77 days	117.7	[59]

RT: room temperature.

**Table 4 materials-15-05961-t004:** Corrosion resistance test results of the geopolymers [37].

Environmental Condition	Time	Phenomena	Compressive Strength (MPa)
100% relative humidity, 20 °C	48 h	Surface unchanged	93.1 ± 5.8
3%NaCl solution	48 h	Surface unchanged	83.8 ± 5.6
1 mol/L HCl	48 h	Sample disintegrated completely and solution changes into yellow	/
1 mol/L NaOH	48 h	Surface changed into black	41.6 ± 4.9

**Table 5 materials-15-05961-t005:** Modification principle of calcium, magnesium, and iron in ASP geopolymers.

No.	Equations	Ref.
(4)	${3\mathrm{Ca}}^{2+}+2{\left[H_{n}{\mathrm{PO}}_{4}\right]}^{n-3}\;\to {\;\mathrm{Ca}}_{3}{{(\mathrm{PO}}_{4})}_{2}{+2\mathrm{nH}}^{+}$	[79]
(5)	${5\mathrm{Ca}}^{2+}+3{\left[H_{n}{\mathrm{PO}}_{4}\right]}^{n-3}{+H}_{2}O\;\to {\;\mathrm{Ca}}_{5}{\left({\mathrm{PO}}_{4}\right)}_{3}\left(\mathrm{OH}\right)+\left(3n+1\right)H^{+}$	[79]
(6)	${\mathrm{Ca}}^{2+}+{\left[H_{n}{\mathrm{PO}}_{4}\right]}^{n-3}{+2H}_{2}O\;\to {\;\mathrm{CaHPO}}_{4}\cdot {2H}_{2}O+\left(n\;-\;1\right)H^{+}$	[79]
(7)	${\mathrm{CaHPO}}_{4}\cdot {2H}_{2}O(\mathrm{Brushite})\overset{180–220\;{}^{\circ}C}{\to }{\mathrm{CaHPO}}_{4}{(\mathrm{Monetite})+2H}_{2}O$	[35]
(8)	${2\mathrm{CaHPO}}_{4}\overset{600–750\;{}^{\circ}C}{\to }{\mathrm{Ca}}_{2}P_{2}O_{7}{+H}_{2}O$	[35]
(9)	${\mathrm{MgO}+\mathrm{XH}}_{2}{\mathrm{PO}}_{4}{+\mathrm{nH}}_{2}O\;\to {\;\mathrm{MgXPO}}_{4}\cdot \left(n+1\right)H_{2}O$	[3]
(10)	$2\mathrm{Al}{\left(H_{2}{\mathrm{PO}}_{4}\right)}_{3}+5\mathrm{MgO}+\left(n+1\right)H_{2}O\;\to {\;2\mathrm{MgHPO}}_{4}\cdot {3H}_{2}O+3\mathrm{MgO}\cdot {\mathrm{Al}}_{2}O_{3}\cdot {2P}_{2}O_{5}\cdot {\mathrm{nH}}_{2}O$	[58]
(11)	${\mathrm{Fe}}_{3}O_{4}{+8H}_{3}{\mathrm{PO}}_{4}\;\to \;\mathrm{Fe}{\left(H_{2}{\mathrm{PO}}_{4}\right)}_{2}+2\mathrm{Fe}{\left(H_{2}{\mathrm{PO}}_{4}\right)}_{3}{+4H}_{2}O$	[37]
(12)	${\mathrm{Fe}}_{3}O_{4}{+4H}_{3}{\mathrm{PO}}_{4}\;\to {\;\mathrm{FeHPO}}_{4}+\mathrm{Fe}{\left(H_{2}{\mathrm{PO}}_{4}\right)}_{3}{+4H}_{2}O$	[37]

**Table 6 materials-15-05961-t006:** The performance parameters of ASP geopolymer porous foams prepared with different foaming agents.

Foaming Agent	Content (%)	Total Porosity (%)	Thermal Conductivity (W/mK)	C S (MPa)	Ref.
Al powder	0.04~0.22	40~83	-	>6	[51]
Limestone	1~5	30.5~32.1	0.133~0.211	>4	[54,87]
Limestone	4	68~70	0.092~0.095	-	
Surfactant	15.7	78.3	-	0.64	[63]
H_2_O_2_	2~4	55~64	-	1.17	[71]

C S: Compressive strength.

## Data Availability

The data presented in this study are available on request from the corresponding author.

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
