# Peer review of "Comprehensive Understanding of Aluminosilicate Phosphate Geopolymers: A Critical Review"

_materials, 2022, doi:10.3390/ma15175961_

Round 1

Reviewer 1 Report

The paper "Comprehensive understanding of aluminosilicate phosphate geopolymers: A critical review" presents an interesting and relevant topic about geopolymers in terms of literature review, it can be considered after further corrections:

(a) The abstract in general is generic and does not provide information on the main databases where the papers studied were collected;

(b) Note that a literature review should be very comprehensive, some relevant topics and more detail on the science and types of geopolymers and activated alkali materials should be provided at the beginning of this paper, some research should be entered and considered, such as: 10.3390/polym13152493; 10.1016/j.cscm.2021.e00723; 10.1016/j.cscm.2021.e00802.

(c) Authors can insert more explanatory figures and other important results, note that there is a special edition in journal open access that can be used and has numerous researches on the topic, such as: https://www.sciencedirect.com/journal /case-studies-in-construction-materials/special-issue/100C1LHWPSX

(d) The total number of references and their scope is limited considering a review paper, which in general has around 100 references, consider the previous comments to expand your conclusions;

(e) Some topics need to be more critical, comparing multiple results with the state of the art of the investigated topic;

(f) A topic on gaps in the literature and problems in researched investigations can be added.

Author Response

Dear reviewer:

According to your valuable opinions, we have adjusted the structure and revised the content of this article manuscript.

Once again, thank you very much for your comments and suggestions.

Best regards

Reviewer 2 Report

The authors in this paler present a critical review pertaining to aluminosilicate phosphate geopolymers. The review is very well prepared, and the structure of the manuscript is in my opinion excellent starting with important description of the formation  mechanisms, following to the properties of these materials and ending up to novel applications e.g. fiber involving materials. They also include a future perspective Paragraph which I believe is very useful for the reader. I could observe only one omission. Even though there is a large number of recent works encompassing geopolymers of the class the authors describe and graphene, the authors have not included any of these studies in this review. I strongly believe that adding a short paragraph reporting new trends involving graphene or grpaphene-based materials would strengthen this review even more and could reach broader research audiences. 

Author Response

(The authors gave the same response as above.)

Reviewer 3 Report

Dear Editors, please find the review report.,

After careful reading of the article "Comprehensive understanding of aluminosilicate phosphate geopolymers: A critical review ", please find the suggestion to improve the paper quality., 

The literature review is quite shallow and the authors should cover the most recent papers as well. 

The conclusion should be more in details. 

Please provide the number of publication versus year on ASP

Please provide the number of patents versus year on ASP

How ASP geopolymer is much better interms of environmental sustainability? authors should justify

Author Response

(The authors gave the same response as above.)

Round 2

Reviewer 1 Report

ok.